# The relationship between anaemia and the use of treated bed nets among pregnant and non-pregnant women in Ghana

Richard Kwame Ansah[1]*, Sampson Tackie[1], Rhodaline Abena Twum[1], Kassim Tawiah[1], Richard Kena Boadi[4], Dorcas Attuabea Addo[3], Samuel Effah-Poku[2], David Delali Zigli[5]

1 Department of Mathematics and Statistics, University of Energy and Natural Resources, Sunyani, Ghana, 2 School of Technology, Christ Apostolic University College, Kumasi, Ghana, 3 Department of Mathematics Education, University of Education, Winneba, Ghana, 4 Department of Mathematics, Kwame Nkrumah University of Science and Technology, Kumasi, Ghana, 5 Department of Mathematical Sciences, University of Mines and Technology, Tarkwa, Ghana

* richard.ansah@uenr.edu.gh

**Data Availability Statement:** The dataset used in this study is included in the supplementary file.

**Funding:** The authors received no specific funding for this work.

## Abstract

Studies have indicated that the risk of malaria, particularly its association with anaemia in pregnant women, increases when treated bed nets are not used. This paper utilizes a statistical mechanical model to investigate whether there is a statistical relationship between the presence or absence of anaemia in pregnant and non-pregnant women and their decision to sleep under treated bed nets. Data from the Ghana Malaria Indicator Survey (GMIS), which includes both rural and urban malaria-endemic areas in Ghana, were employed in this study. A total of 2,434 women, comprising 215 pregnant and 2,219 non-pregnant participants, were involved. Among these, 4.76% of the pregnant and anaemic women and 45.89% of the non-pregnant and anaemic women slept under treated bed nets, while 0.86% of the pregnant and anaemic and 6.82% of the non-pregnant and anaemic women did not. The findings revealed that, in the absence of social interaction, non-anaemic pregnant women have a lower prevalence of choosing to use bed nets compared to their anaemic counterparts. Additionally, non-pregnant anaemic women showed a positive private incentive (30.87%) to use treated bed nets, implying a positive correlation between anaemia and the choice to sleep in a treated bed net. Furthermore, the study demonstrated that both pregnancy and anaemia status have a relationship with the use of treated bed nets in Ghana, especially when social interactions are considered. The interaction strength between non-pregnant and anaemic women interacting with each other shows a negative estimate (-1.49%), implying that there is no rewarding effect from imitation. These insights are crucial for malaria prevention and control programs, emphasizing the need for targeted interventions to enhance the use of treated bed nets among both pregnant and non-pregnant women in Ghana's malaria-endemic regions.

**Competing interests:** The author declares that he has no competing interests.

## Introduction

Anaemia refers to a medical condition characterized by below-normal levels of red blood cells or haemoglobin concentration in the blood, leading to a reduced capacity to carry oxygen and presenting symptoms like fatigue, weakness, dizziness, and breathlessness. Its causes include nutritional deficiencies, haemoglobinopathies, and infectious diseases such as malaria etc. Anaemia is a significant global public health concern, particularly affecting young children and pregnant women. According to World Health Organization (WHO), anaemia affects approximately 42% of children under the age of five and 40% of pregnant women worldwide [1]. Anaemia poses a significant global public health concern that has adverse effects on the social and economic progress. In developing nations, approximately 50% of women and children suffer from anaemia, with Southern and Central Asia and specific African regions having the highest prevalence rates [2].

Anaemia is typically identified by conducting a blood test. To diagnose anaemia among women aged 15–49 and children between 6 months and 5 years, the Demographic Health Surveys (DHS) Program utilizes the HemoCue blood haemoglobin testing system, which involves either finger prick or heel prick blood testing. Participation in the testing is voluntary, and individuals are given their anaemia test results immediately, along with guidance on how to prevent anaemia [2].

Insecticide-treated bed nets (TBNs) are personal protectives nets that are used to cover the upper parts of beds so that one can safely sleep under them to serve as a measure that lowers the incidence of malaria, severe sickness, and mortality [3]. TBNs have demonstrated a reduction of about 20% in the overall death rate among children under the age of five, according to community-wide studies conducted in various African contexts. These bed nets provide a protective barrier, encircling individuals sleeping beneath them [3]. Additionally, TBNs are much more protective than untreated ones. TBNs are essential and demonstrate safe keeping to curtail malaria infection and acute diseases in endemic regions. In numerous African regions, trials in communities have shown evidence that sleeping under TBNs can reduce deaths by over twenty percent [3]. The insecticides in TBNs not only eliminate mosquitoes and other insects but also act as a deterrent, reducing the number of mosquitoes that enter homes and attempt to feed on sleeping individuals. Moreover, if more than fifty percent of people in a community sleep under treated bed nets, it could also decrease the lifespan of all forms of mosquitoes [3]. TBNs have been proven to be a major factor in reducing the spread of malaria [3].

Malaria is characterized by the World Health Organization (WHO) as a serious febrile illness, resulting from the Plasmodium parasite, which can be transmitted via blood transfusion, congenitally, or more commonly, through the bite of an infected female Anopheles mosquito [4]. Symptoms such as fever, headache, and chills typically emerge 10–15 days following a mosquito bite from an infected insect. These manifestations can be subtle and difficult to identify as malaria. In regions where malaria is prevalent, individuals with weaker immune defences may experience asymptomatic infections, showing no noticeable symptoms [4].

WHO advises everyone who suspects they may have malaria to be tested right away. If treatment for Plasmodium falciparum malaria is not received within a day, the infection may worsen and result in death. While severe anaemia, respiratory distress, or cerebral malaria are common in children, severe malaria can lead to multi-organ failure in adults. Other Plasmodium species that cause malaria in humans can sometimes result in serious sickness, including potentially fatal conditions. Tests that identify the presence of the parasites causing malaria can be used to diagnose the illness. Rapid tests for diagnosis and microscopic analysis of blood smears are the two primary test kinds. By using diagnostic tests, medical professionals can differentiate between malaria and other febrile disease causes, allowing for more targeted therapy [4].

Malaria has consistently been a high burden disease in Africa and a majority of developing countries. As a high-burden disease, it has a negative impact on the economies of already struggling developing countries, the majority of which are in Africa [5, 6]. Effective surveillance is necessary at every stage of malaria control and elimination. Human and non-human resources are the components required for an effective surveillance system, which includes technologies, procedures, people, and structures [7].

Some studies have emphasized the challenges of inadequate institutions, lack of surveillance technology, noncompliance of program officers with current processes, general resource shortage, and substandard capacity of program officers for malaria surveillance [8]. Consequently, efforts to eradicate malaria have proven elusive with planning and intervention operations highly tough [9]. Up until now, one of the technologies or instruments used to reduce mosquito bites is the insecticide-treated bed net. In malaria-endemic nations, TBNs have been shown to be useful in the prevention and control of malaria. Yet, home usage varies and can significantly impact the benefits of TBNs as a malaria transmission prevention technique [10, 11].

[12] examined malaria in pregnant women, covering aspects like epidemiology, pathology, clinical symptoms, diagnosis, and treatment. The study showed that various interventions such as TBNs, intermittent preventive treatment, and case management of malaria and its associated anaemia can help prevent malaria. The WHO suggests that the distribution of TBNs should cover everyone in areas where malaria is common, not just vulnerable groups like pregnant women and young children. In many African countries, including Ghana, TBNs are given out for free during mass campaigns, and pregnant women receive them for free during antenatal and postnatal care. Despite these efforts, malaria remains a significant problem in Africa, with fluctuating death rates over the years. In 2021, an estimated 40 million pregnancies with moderate to high malaria occurred in 38 African countries, and 13.3 million pregnancies were at risk of contracting malaria during pregnancy [1].

TBNs are highly effective against late-night biting mosquitoes and provide significant personal protection even when damaged. Distributing or retreating them for free is a cost-effective method of protecting vulnerable groups in malaria-endemic communities. High coverage in the community reduces mosquito survival and sporozoite-positive mosquitoes. Although long-term use of treated nets may reduce anti-malaria antibody levels in children, nets still reduce anaemia and mortality rates. Alternative fabric treatments should be developed in case of stronger forms of resistance [13]. Pregnant women are at a higher risk of experiencing severe malaria, which can lead to negative outcomes such as severe anaemia maternal death, low birth weight, and stillbirth.

The utilization of TBNs is influenced by a variety of individual, household, and community-level factors, such as age, gender, number of bedrooms, wealth status of the household, prevalence of malaria, location of residence, and region of abode. By focusing on these factors, the use of TBNs can be improved, which would enhance malaria prevention efforts [14]. [15] analysed the effects of indoor residual spraying (IRS) on malaria outcomes in some eastern parts of Africa. This region has high and persistent malaria transmission, with moderate use of TBNs (55–65% of individuals used a net the night before). [15] revealed that IRS are not that effective like TBNs. While TBNs and IRS have been widely implemented in Sub-Saharan Africa for malaria prevention, there is a gap in study regarding the benefits of IRS in areas with moderate to high TBN coverage.

[16] showed that TBNs decreased the occurrence of malaria by seventy-four percent (74%) and delayed the first occurrence of parasitaemia. In addition, the incidence of clinical malaria and anaemia decreased by sixty percent (60%). The protective effect of TBNs was most significant in infants under three months old and was consistent in older infants and one-year-old

children. The study concluded that the use of TBNs greatly reduced malaria exposure and related health issues.

Malaria is associated with high anaemia level [17]. Several studies have shown that TBNs prevent malaria infection in people [18, 19]. We can therefore imply that TBNs prevent anaemia through the prevention of malaria. In pregnant women, it is a key factor for a healthier maternal life [20–22]. Again, [23–25] showed that sleeping in TBNs has direct and indirect effects on haemoglobin levels in humans, thereby playing a major role in the prevention of anaemia. Interventions are mostly needed to achieve targeted anaemia prevention in all areas [26].

In endemic areas such as Ghana, the use of TBNs are one of the proven strategies for reducing malaria and its associated anaemia vis-à-vis its fatalities. As a result, women (pregnant and non-pregnant) are most likely to be advised to use TBNs [27]. This notwithstanding, although most women (pregnant and non-pregnant) (around eighty-five percent (85%)) have TBNs [27], a greater proportion do not use them [28, 29]. Several factors, including pregnancy status, health status (i.e anaemia level) etc., have been found to influence the usage of TBNs among women in Ghana [20, 28].

In this study, we aim to understand how complex interactions among women within a reference group influence their collective behaviour [30–32]. The phenomenon of self-organization—a type of collective behaviour—has been observed across various systems, including biological, ecological, and socio-economic contexts [33, 34]. In such systems, even slight modifications to the socio-economic structure can precipitate substantial changes in the collective behaviour of a large group of individuals. For instance, the introduction of a small immigrant population might alter language pronunciation patterns, while proactive measures by authorities have been shown to significantly reduce crime rates [35, 36].

The concept of phase transition refers to a sudden change in macroscopic behaviour caused by alterations in the interactions among its components. This concept originates from the field of statistical mechanics. Certain spin models, which aim to explain ferromagnetism, have demonstrated the existence of phase transitions [37, 38]. One such spin model, known as the mean-field Ising model proposed in [37], belongs to this category. It has been widely applied in various disciplines, including social sciences, finance, chemistry, and ecology [39–42]. Notably, a multispecies version of the mean-field Ising model, which is particularly relevant to the study of magnetism in anisotropic materials, has emerged from practical applications [43]. In the realm of social sciences, studies such as [44, 45] have utilized this model for analysing social phenomena. The aforementioned findings, along with the work presented in [30–32], highlight the significance of incorporating statistical mechanical models into social science study. Such integration can provide valuable insights into the pivotal role of social interactions in shaping social outcomes.

The multi-population Curie-Weiss model provides a framework for studying binary discrete choices, such as deciding between two options, like staying in school or dropping out, or choosing whether to use medicated mosquito nets while sleeping at night. These choices are influenced by individuals' socio-economic environments. The model posits that people with similar socio-economic attributes tend to exhibit similar behaviour, whereas those with differing attributes may behave differently [45]. Previous research, as highlighted in works by [44, 45], has delved into a version of the Curie-Weiss model tailored for multiple groups. This approach keeps the ratio of individuals in each subgroup stable, irrespective of the overall population count. For example, [45] study employed this framework to analyse factors linked to suicidal behaviour and marriage trends in Italy, with a focus on residence as a key socio-economic factor. Similarly, [46] research investigated the impact of socio-economic factors, including gender and residence, on educational outcomes in five selected African developing

nations. For a broader comprehension of how statistical mechanics models are applied in social sciences, [30–32] provide valuable insights.

This study investigated the statistical relationship between the use of insecticide-treated bed nets, anaemia, and pregnancy status. We employed a statistical mechanical model, specifically the Curie-Weiss model with multiple populations to analyse whether these characteristics, anaemia and pregnancy status, have a relationship in determining a woman's preference for sleeping in TBNs. We adapted the partial least squares estimation technique to determine the parameters of the Curie-Weiss model with multiple populations. The results will be crucial for the general public, as they will assist the Ghanaian government, the Ghana Health Service, and all stakeholders in strategizing effectively and revising their campaigns to encourage all women to use Treated Bed Nets (TBNs). It is imperative to note that campaigns and strategies aimed at encouraging all women, irrespective of their status, to sleep under TBNs are essential. This effort is critical to ensure the entire family and household adopt the use of TBNs, given that women play a pivotal role and serve as influential role models in making decisions of this nature.

## Methods

### Study design and data

This study presented an analytical cross-sectional study that focused on women across the original 10 regions of Ghana. The investigation was based on data from the 2019 Ghana Malaria Indicator Survey (GMIS), which, in turn, relied on the framework of the 2010 Population and Housing Census (PHC) implemented by the Ghana Statistical Service (GSS). It is important to note that Ghana underwent a regional reorganization in 2019, increasing its regions from 10 to 16, thereby incorporating an additional 260 districts and municipalities. Despite this change, the 2019 GMIS did not include these new regions due to their recent formation; thus, the survey was conducted within the boundaries of the 10 regions defined in the 2010 PHC. The GMIS aimed to collect comprehensive data on malaria and its impact, offering valuable information not just for the country as a whole but also for its specific urban and rural zones, including the ten regions identified in the 2010 PHC: Western, Central, Greater Accra, Volta, Eastern, Ashanti, Brong Ahafo, Northern, Upper East, and Upper West. The survey's sampling was based on all census enumeration areas (EAs) from the 2010 PHC as provided by the GSS [47, 48].

The 2019 GMIS study utilized a two-stage sampling process, segmented into 20 distinct groups based on the urban-rural classification in each region. Initially, 200 Enumeration Areas (EAs) were selected, split into 97 urban and 103 rural EAs, following a method of independent choice and size consideration in each group, as elaborated in [47, 48]. In the subsequent stage, 30 households were chosen from each group, cumulatively amounting to 6,000 households. The selection framework within each group was arranged for implicit stratification and proportionate distribution at more granular administrative levels. For an in-depth understanding of the 2019 GMIS sampling, inclusive of data gathering techniques, instruments, and quality assurance, the full report is available in [47, 48]. This study's analysis was particularly focused on the data from 2,434 women, aged between 15 and 49 years.

### Study variables

The study focused on women's use of treated bed nets as the outcome variable. Usage was categorized as either sleeping under a treated bed net (coded as +1) or not (coded as -1). The covariates included in this study were pregnancy status (pregnant or non-pregnant) and anaemia status (anaemic or non-anaemic).

### Ethics statement

This research was based on data extracted from the 2019 Ghana Malaria Indicator Survey (GMIS). The authors received authorization to use the data from MEASURE DHS/ICF International. The procedures of the Malaria Indicator Survey (MIS) Programme align with established guidelines for protecting the confidentiality of respondents. ICF International is responsible for ensuring that the survey complies with the Human Subjects Protection Act as set by the U.S. Department of Health and Human Services. The MIS project had previously obtained all necessary ethical clearances before the execution of the survey, thus obviating the need for additional approvals for this study [48]. More information on data and ethical considerations can be found at http://goo.gl/ny8T6X.

## Statistical analysis

### The Curie-Weiss Hamiltonian

For a set of coded decisions, $\delta = (\delta_1, \delta_2, \cdots, \delta_D)$, the Curie-Weiss Hamiltonian at inverse temperature $K_{z,n}$ and an external magnetic field $p_z$ are defined as follows:

$$H_D(\delta) = \frac{1}{2D}\sum_{z,n=1}^{D} K_{z,n}\delta_z\delta_n + \sum_{z=1}^{D} p_z\delta_z \tag{0.1}$$

where

$$\delta_z = \begin{cases} +1, & \text{if individual } z \text{ sleeps in TBN,} \\ -1, & \text{if individual } z \text{ does not sleep in TBN .} \end{cases}$$

The Curie-Weiss Hamiltonian is divided into two parts: the interaction component, $K_{z,n}$, which represents the interaction strength between individuals $z$ and $n$, and the external field of influence, $p_z$. If $K_{z,n}$ is positive, it indicates that conformity is rewarded, if $K_{z,n}$ is negative, it indicates that conformity is not rewarded. Spins pointing up are favoured when the external field $p_z$ is directed upwards, while spins pointing down are favoured when the field is directed downwards [49].

In this work, the Multi-population Curie-Weiss model for discrete choice with social interactions will be used as a benchmark model.

### Multipopulation Curie-Weiss model

Our main premise was that women with similar characteristics exhibited similar behaviour, whereas women with diverse attributes displayed diverse behaviours. This premise played a crucial role in redefining the parameters of the Curie-Weiss Hamiltonian as shown in Eq (0.1). Consequently, our primary objective was to determine a suitable parameterization for the interaction coefficient $K_{z,n}$ and develop a systematic approach for estimating the model's parameters using data. In line with our discrete choice model, each individual $z$ was assigned a set of $w$ characteristics.

$$\varphi_z = (\varphi_z^1, \varphi_z^2, \cdots, \varphi_z^w), \text{with } \varphi_z^j \in \{0, 1\}. \tag{0.2}$$

Consider the example where the parameters of interest are pregnancy status $\varphi_z^1$ and anaemia status $\varphi_z^2$ with

$$\varphi_z^1 = \begin{cases} 1, & \text{if woman } z \text{ is pregnant} \\ 0, & \text{if woman } z \text{ is non-pregnant} \end{cases} \tag{0.3}$$

**Table 1. Population classification based on attributes.**

| Attributes | Anaemic | Non-anaemic |
|---|---|---|
| Pregnant | 137 | 78 |
| non-pregnant | 1283 | 936 |

and

$$\varphi_z^2 = \begin{cases} 1, & \text{if woman } z \text{ is anaemic} \\ 0, & \text{if woman } z \text{ is non-anaemic.} \end{cases} \tag{0.4}$$

## Estimation

The least squares method is used to calculate the model parameters. As a result, we must identify the parameter settings that minimizes

$$\sum_a [\bar{n}^a - \tanh(U_a)]^2 \tag{0.5}$$

where $\bar{n}^a$ is the average choice of group $a$. Because $\tanh(U_a)$ is non-linear, the computations will take an extremely long time, see [46]. In the interaction scenario, the independent variables are correlated. As a result, the least squares method is rendered ineffective. In that case, the partial least squares estimation method will be utilized. We utilized MATLAB software version R2016a for carrying out our analyses and obtaining results.

This study utilized secondary data from the Ghana Malaria Indicator Survey (GMIS), which included information on the pregnancy and anaemia status of 2,434 women in Ghana. It also examined their use of treated bed nets. The participants comprised 215 pregnant and 2,219 non-pregnant women. Among them, 4.76% of the pregnant and anaemic women, and 45.89% of the non-pregnant and anaemic women, had slept under treated bed nets. Conversely, 0.86% of the pregnant and anaemic women, and 6.82% of the non-pregnant and anaemic women, had not used treated bed nets. Approximately half (49%) of the participants were between the ages of 25 and 34 years. Here we looked at women who had been partitioned according to two attributes $\varphi_z^1$ and $\varphi_z^2$ representing pregnancy and anaemia status. Table 1 below shows the partition of the population into four subsection: pregnant and anaemic, pregnant and non-anaemic, non-pregnant and anaemic, and non-pregnant and non-anaemic.

**Table 2. Women that sleep in treated bed nets.**

| Attributes | Anaemic | Non-anaemic |
|---|---|---|
| Pregnant | 116 | 71 |
| Non-pregnant | 1117 | 789 |

**Table 3. Women that do not sleep in treated bed nets.**

| Attributes | Anaemic | Non-anaemic |
|---|---|---|
| Pregnant | 21 | 7 |
| Non-pregnant | 166 | 147 |

**Table 4. Attributes of the population.**

| | Attribute | | | |
| | Pregnancy status ($\varphi_z^1$) | | Anaemia status($\varphi_z^2$) | |
| Cases | Pregnant | Non-pregnant | Anaemic | Non-anaemic |
|---|---|---|---|---|
| 1 | 0 | 1 | 1 | 0 |
| 2 | 1 | 0 | 0 | 1 |
| 3 | 0 | 1 | 0 | 1 |
| 4 | 1 | 0 | 1 | 0 |

Tables 2 and 3 below display the distribution of the population by their use of treated bed nets, segmented according to various attributes.

We got four groups that were indexed with this type of partitioning by $a = 1, \cdots, 4$.

In particular,

$a = 1$ represents the group of pregnant women that are anaemic,

$a = 2$ represents the group of pregnant women that are non-anaemic,

$a = 3$ represents the group of non-pregnant women that are anaemic,

$a = 4$ represents the group of non-pregnant women that are non-anaemic.

We then computed the values of the attributes $\varphi_z^1$ and $\varphi_z^2$ representing pregnancy and anaemia status depending on the group a woman belongs to, for the four cases generated in Table 4 below. The values assigned to the attributes in each of the four cases described the relevance of that attribute to the private incentive part of the Hamiltonian. For example, in Case 1, the attribute of being a pregnant woman did not contribute to the private incentive of a group, whereas the attribute of being a non-pregnant woman did contribute to the group's private incentive.

## Results

### Non interacting case

When the sum of the $\tau_j$'s of the private incentives is positive, that group will make a decision that favours choosing to sleep in a treated bed net, and when the sum is negative, that group will make a decision of not sleeping in a treated bed net. In Table 5, the estimates have been compiled. The sum of the $\tau_j$'s for cases 1, 3 and 4 yields a positive value, suggesting that those

**Table 5. Estimates for the non-interacting model.**

| Cases | Estimation of Parameters |
|---|---|
| Case 1: non-pregnant women that are anaemic | $\tau_1 = 0.3087$ |
| | $\tau_2 = 0.0776$ |
| | $\tau_0 = -0.0061$ |
| Case 2: Pregnant women that are non-anaemic | $\tau_1 = -0.3087$ |
| | $\tau_2 = -0.0776$ |
| | $\tau_0 = 0.3802$ |
| Case 3: non-pregnant women that are non-anaemic | $\tau_1 = 0.3087$ |
| | $\tau_2 = -0.0776$ |
| | $\tau_0 = 0.0715$ |
| Case 4: Pregnant women that are anaemic | $\tau_1 = -0.3087$ |
| | $\tau_2 = 0.0776$ |
| | $\tau_0 = 0.3026$ |

who are taken into account in these cases will decide to sleep in a treated bed net. The fact that Case 2 sum of $\tau_j$'s is negative suggests that when there is no social interaction, pregnant women that are non-anaemic will not decide to sleep in a treated bed net. Cases 1 and 3 have positive estimates for the private incentive of pregnancy status $\tau_1$. This implies that being non-pregnant has a positive association or correlation on the choice of sleeping in a treated bed net. Case 1 has positive estimates for the private incentive of anaemia status $\tau_2$. This implies that being anaemic has a positive correlation on the choice of sleeping in a treated bed net. The base private incentive $\tau_0$ has positive values for three cases, signifying that individuals will prefer the choice of sleeping in a treated bed net. Cases 2, 3, and 4 will opt for sleeping in a TBN since they have positive base private incentives.

## Interacting case

The utility function of the interactive model incorporates both social and private incentives. When the social incentive $K_{a,b}$ is positive, individuals in groups $a$ and $b$ prefer to imitate each other. Conversely, when $K_{a,b}$ is negative, people in these groups tend to make choices with different signs, indicating that conformity is not desirable in such circumstances. Note that Table 6 shows a negative estimate for $K_{11}$, which represents the interaction strength between non-pregnant and anaemic women interacting with each other. This implies that there is no rewarding effect from imitation. On the other hand, $K_{33}$ represents the interaction strength of non-pregnant and non-anaemic women interacting with each other. The estimate for $K_{33}$ is positive, this means the imitation is rewarded. If the sum of $\tau_j$'s representing the private incentives of a group is positive, then the women in that group will choose to sleep under a treated bed net. Conversely, if the sum is negative, the women in the group will choose not to sleep under a treated bed net. $\tau_1$ and $\tau_2$ are positive signifying that the pregnancy and anaemia status have an association or positive correlation on sleeping in a treated bed net in Ghana, if social interaction is present. The base private incentive $\tau_0$ is zero, which means that in a situation where social

**Table 6. Estimates for the interacting model.**

| Parameter | Estimate |
|---|---|
| $K_{11}$ | -0.0149 |
| $K_{12}$ | -0.0100 |
| $K_{13}$ | -0.1490 |
| $K_{14}$ | -0.1006 |
| $K_{21}$ | -0.0229 |
| $K_{22}$ | -0.0155 |
| $K_{23}$ | -0.2296 |
| $K_{24}$ | -0.1550 |
| $K_{31}$ | 0.0244 |
| $K_{32}$ | 0.0164 |
| $K_{33}$ | 0.2442 |
| $K_{34}$ | 0.1649 |
| $K_{41}$ | 0.0134 |
| $K_{42}$ | 0.0090 |
| $K_{43}$ | 0.1344 |
| $K_{44}$ | 0.0908 |
| $\tau_1$ | 0.9691 |
| $\tau_2$ | 0.2437 |
| $\tau_0$ | 0 |

**Table 7. Variance of $n^a$ and $U_a$ explained by the latent vectors and RMSEP for Ghana.**

| Latent Vectors | Percentage of Explained Variances for $n^a$ | Cumulative Percentage of Explained Variances for $n^a$ | Percentage of Explained Variances for $U_a$ | Cumulative Percentage of Explained Variances for $U_a$ | RMSEP for $n^a$ | RMSEP for $U_a$ |
|---|---|---|---|---|---|---|
| 1 | 45.75 | 45.75 | 97.34 | 97.34 | 0.8175 | 0.1625 |
| 2 | 8.45 | 54.2 | 2.66 | 100 | 0.6022 | 0.0265 |
| 3 | 45.8 | 100 | 0 | 100 | 0.5533 | 0 |

interaction is present, women considered in these cases will rely on their pregnancy or anaemia status(anaemic or non-anaemic) to make a choice towards sleeping in a treated bed net.

## Model diagnostics and validation

In this section, we found the goodness of fit of our interacting model to the data. To estimate the parameters of the model proposed in [46], the researchers employed the Partial Least Squares (PLS) method.

These results indicated that the interacting model proposed in [46] fit the data well, with a high level of variance explained by the latent vectors. This demonstrated the efficacy of the PLS method in predicting the dependent variable based on the independent variables. The results also suggested that the model could be used for modelling or prediction with reasonable accuracy.

This approach involved predicting the dependent variable $U_a$ by using the independent variable $n^a$. Through PLS, the independent variables were transformed into orthogonal factors, or latent vectors, which were then ranked based on their eigenvalues in descending order. The number of latent vectors employed was determined by their capacity to clarify the covariance that exists between the independent and dependent variables [50].

Table 7 presents the variance explained by the latent vectors employed in estimating the independent variables ($n^a$) and dependent variable ($U_a$), along with their respective root mean square error of prediction (RMSEP). The table indicates that the first three latent vectors are accountable for 100% of the variance in both independent and dependent variables. The variances explained by the dependent variable are sufficient for either modelling or prediction purposes. Additionally, for all cases, the RMSEP for $n^a$ and $U_a$ decreases as the number of latent vectors increases.

## Discussions

This paper examines the statistical relationships among the use of treated bed nets, the anaemia status (anaemic or non-anaemic), and the pregnancy status (pregnant or non-pregnant) of women in Ghana. The study carries significant implications for communities, enabling the Ghanaian government, the Ghana Health Service, and other stakeholders to tailor their strategies and campaigns more effectively to promote the use of insecticide-treated nets (ITNs) among all women. It is important to note, however, that this study, being primarily an analytical cross-sectional survey, highlights statistical relationships rather than direct cause-and-effect links. Additionally, since the study relied on self-reported data, there was potential for recall bias and a tendency for respondents to provide socially desirable responses.

We focused significantly on understanding the interplay between private incentives, such as personal health benefits, and social incentives, including community acceptance. This detailed approach mirrors the research conducted by the Johns Hopkins Center for Communication Programs [51]. Their study went a step further by highlighting the critical importance

of factors, such as knowledge and attitudes, in the design of effective malaria prevention programs.

Additionally, our observations are consistent with insights from a study conducted in Madagascar [52]. This study highlighted the temporary effectiveness of private incentives in altering health behaviours. It emphasized a crucial finding: although these incentives might initially motivate behavioural changes, such as adopting insecticide treated bed nets, their impact tends to diminish over time. This suggests a need for more sustainable strategies.

Furthermore, an insightful study from Ghana has brought to the forefront the influence of socio-economic factors on health-related decision-making [53]. This study echoed the insights of our study by demonstrating how social factors, such as wealth, social status, and access to resources, significantly shape individuals' decisions regarding health practices, including the usage of treated bed nets.

These studies collectively contributed to a more comprehensive understanding of the multifaceted nature of health behaviour decisions. They particularly highlighted the complexity involved in malaria prevention strategies. Emphasizing that effective interventions must consider a wide range of factors, these include privative incentives as well as social incentives to ensure the widespread and sustained use of insecticide-treated bed nets.

Personal health conditions, particularly pregnancy and anaemia status, have a statistical relationship with a woman's decision to use treated bed nets. This highlights the need for health education to specifically target these groups, emphasizing the benefits of bed net usage. Additionally, social dynamics play a role in the decision to use bed nets. Positive social incentives have been shown to increase bed net usage among non-pregnant and non-anaemic women, while negative social incentives may have the opposite effect. This underscores the potential effectiveness of community-based initiatives and peer support in promoting bed net use. Generally, there is a positive base private incentive, indicating a general inclination towards using treated bed nets. However, in scenarios involving social interaction, decisions tend to be more heavily influenced by the social environment.

Based on the findings of this study, several recommendations for future research have been proposed. Given that personal health conditions, such as pregnancy and anaemia status, significantly impact the usage of treated bed nets (TBNs), it is suggested that future studies focus on tailoring health education and intervention programs specifically for these groups. This could involve assessing the effectiveness of targeted messaging and intervention strategies. Additionally, with socio-economic factors like wealth, social status, and access to resources playing a crucial role in health-related decisions, it would be valuable for future research to explore in-depth how these factors influence the use of TBNs. This exploration might include examining the affordability of TBNs and the accessibility of malaria prevention resources across various socio-economic contexts.

## Conclusion

This study has provided significant insights into the behavioural patterns influencing the use of treated bed nets among women in malaria-endemic areas of Ghana. Our analysis, grounded in a statistical mechanical model and leveraging data from the Ghana Malaria Indicator Survey, has highlighted several critical findings. Firstly, it is evident that there is a statistical relationship between a woman's pregnancy and anaemia status and her decision to use treated bed nets. Pregnant women with anaemia have a higher prevalence of choosing to use bed nets compared to their non-anaemic counterparts, underscoring the role of perceived health risks in motivating preventive measures.

Moreover, our study has revealed a nuanced interaction among non-pregnant anaemic women. Despite a positive private incentive to use treated bed nets observed in this group, the interaction strength indicates a lack of a rewarding effect from imitation, suggesting that other factors might play a more significant role in influencing their decision.

The disparity in the usage of treated bed nets, as demonstrated by the low percentages among certain groups, particularly pregnant and non-anaemic women, calls for a more targeted approach in malaria prevention campaigns. These campaigns should not only emphasize the heightened risk of malaria and its severe implications, such as high anaemia during pregnancy, but also tailor their messages to address the specific concerns and perceptions of different demographic groups.

Additionally, the findings suggest that beyond the provision of treated bed nets, there is a crucial need for educational programs that effectively communicate the benefits and necessity of using these nets, especially in rural and urban malaria-endemic areas. Such interventions should be designed to overcome barriers related to social interactions and personal health beliefs.

In conclusion, this study underscores the complexity of health behaviour, particularly in the context of malaria prevention. It calls for a multifaceted approach in public health strategies, one that not only provides necessary preventive tools like treated bed nets but also invests in community-based education and tailored communication strategies. By addressing these aspects, malaria control programs in Ghana and similar regions can enhance the effectiveness of their interventions, ultimately contributing to the reduction of malaria incidence and improving maternal health outcomes.

## Supporting information

**S1 File. Supporting information on the dataset.**
(XLSX)

**S2 File. Supplementary information on the methodology.**
(ZIP)

**S1 Checklist. *PLOS ONE* clinical studies checklist.**
(DOCX)

## Author Contributions

**Conceptualization:** Richard Kwame Ansah, Kassim Tawiah, Richard Kena Boadi, Samuel Effah-Poku.

**Data curation:** Rhodaline Abena Twum.

**Formal analysis:** Richard Kwame Ansah.

**Investigation:** Kassim Tawiah, Richard Kena Boadi.

**Methodology:** Richard Kwame Ansah, Sampson Tackie, Kassim Tawiah.

**Project administration:** Samuel Effah-Poku.

**Resources:** Rhodaline Abena Twum, David Delali Zigli.

**Software:** Richard Kwame Ansah, Sampson Tackie, David Delali Zigli.

**Supervision:** Samuel Effah-Poku.

**Validation:** Sampson Tackie, Dorcas Attuabea Addo, David Delali Zigli.

**Visualization:** Kassim Tawiah.

**Writing – original draft:** Richard Kwame Ansah.

**Writing – review & editing:** Sampson Tackie, Rhodaline Abena Twum, Dorcas Attuabea Addo.

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
