## [Decision Letter · Decision Letter 0]

30 Oct 2023

PONE-D-23-14924Does the Anaemic Level of Pregnant and Non- Pregnant Women Influence their Choice of Sleeping in a Treated Bed Net?PLOS ONE

Dear Dr. Ansah

Thank you for submitting your manuscript to PLOS ONE. After careful consideration, we feel that it has merit but does not fully meet PLOS ONE’s publication criteria as it currently stands. Therefore, we invite you to submit a revised version of the manuscript that addresses the points raised during the review process.

We look forward to receiving your revised manuscript.

Kind regards,

Ibrahim Sebutu Bello, MBBS, MPH, FMCGP

Academic Editor

PLOS ONE

Journal Requirements:

5. Please update your submission to use the PLOS LaTeX template. The template and more information on our requirements for LaTeX submissions can be found at http://journals.plos.org/plosone/s/latex.

Additional Editor Comments (if provided):

The author need to address all the issues raised by the two reviewers

Reviewers' comments:

Reviewer's Responses to Questions

**Comments to the Author**

1. Is the manuscript technically sound, and do the data support the conclusions?

Reviewer #1: Yes

Reviewer #2: No

2. Has the statistical analysis been performed appropriately and rigorously? 

Reviewer #1: I Don't Know

Reviewer #2: No

3. Have the authors made all data underlying the findings in their manuscript fully available?

Reviewer #1: Yes

Reviewer #2: Yes

4. Is the manuscript presented in an intelligible fashion and written in standard English?

Reviewer #1: Yes

Reviewer #2: Yes

5. Review Comments to the Author

Reviewer #1: Revise your title of study, this should not be a question.

Please add statistics significance in result part of abstract

Key words should be more specific

Public health significance may be highlighted in the introduction part

Conclusion needs to be revised.

Please separate discussion part from result.

Revise references Journal title as per NLM catalogue.

Correct English grammatical error

Reviewer #2: The study addresses a topic of great interest and relevance, especially in countries where malaria is endemic, such as in Africa. However, there is a clear need for substantial revisions to improve the structure of the paragraphs in order to provide greater clarity and fluency to the topic. This can be achieved by reducing repetitions of unaddressed objects and emphasizing the detailing of significant points.

Furthermore, it is important to note that the methodology of the work, while providing a detailed analysis of the statistical model formulas used, lacks crucial information. Such information includes the study design, the contexts in which the research was conducted, eligibility criteria, a description of data sources, the variables analyzed, and the treatments applied. These aspects are fundamental in epidemiological articles.

Finally, the discussion section, which is not present in the article, needs to be incorporated. The absence of this essential element for the understanding and interpretation of the results represents a gap in the study.

To enhance the quality of the article and meet epidemiological research standards, it is recommended to use the tools recommended by the journal Plos One, such as STROBE.

Keywords: partial least squares; Anaemic level; Social interaction; Treated bed net

Due to the importance of keywords for article discovery, it is suggested to review the proposed terms, as only "Social Interaction" is present in the DECS and MESH indexes.

Regarding the description of where the data can be obtained in complete sentences, a link is provided, but the identifiers (IDs) or links to the corresponding data set are not mentioned. Including this information is recommended to enable location and access.

INTRODUCTION

The introduction of the article presents epidemiological evidence and data relevant to the research context. However, it needs to be restructured to make it more logical, cohesive, and aligned with academic writing standards. Furthermore, the arguments, which sometimes appear disjointed or insufficiently justified, need to be qualified.

The following reorganization of the introduction is suggested:

1. Definitions: Initially, it is essential to clearly present and define all objects of study. This includes treated bed nets, malaria, and anemia. Establishing these definitions at the beginning of the work will provide a solid foundation for the reader's understanding.

2. Risk Factors and Consequences: Next, I recommend presenting the consequences and associated factors related to malaria, anemia, and the non-use of bed nets in a structured manner. Although these elements have already been mentioned in various paragraphs of the introduction, structuring this information will avoid repetitions and enhance the clarity and cohesion of the text.

3. Outcomes and Impact: It is relevant to present, in a fluid and structured manner, the statistics related to the outcomes and consequences addressed throughout the work. This section should connect the outcomes presented in the article, providing an overview.

4. Justification: It is advisable to include a justification section that demonstrates the importance of the research conducted. This involves presenting pre-existing hypotheses and explaining why the research is relevant and necessary.

Regarding anemia, it is important for the introduction to provide a clearer explanation of how anemia relates to the article's theme and why it is relevant to study it. This will ensure that readers understand the importance of this variable in the research context.

Finally, it is advisable to remove the paragraph that presents the structure of the article, as this is already widely known and should follow the journal's guidelines. The introduction should focus on establishing the context and relevance of the study.

METHOD

It is highly recommended that the authors follow the STROBE guidelines, as suggested by Plos One. These guidelines are especially relevant for epidemiological articles, as they provide a framework to ensure the understanding and reproducibility of research in different contexts. Certainly, this will contribute to the quality and clarity of the article.

A clear description of the research design adopted in the study should be included. It appears to be an analytical cross-sectional study, but this information should be explicitly provided. Additionally, it is essential to provide the study's context, including the population size, characteristics of the country where the research was conducted, and any other details that help contextualize the work.

Detailed information about participant eligibility criteria, exposures, outcomes, and any other relevant variables must be provided. These descriptions should be clear and included in the methods section so that readers can fully understand the scope of the study.

The description of whether the data was anonymized or not should be in the method section, not in the results. This will contribute to a clearer organization of the article.

While it is evident that the authors have experience in statistical analysis, it is important to consider that readers may include healthcare professionals, managers, and academics who are not experts in the field. Therefore, it is recommended that mathematical formulas be provided as supplementary material, and that the authors describe the motivations behind their methodological choices, explaining how the results were processed. Additionally, creating a flowchart to present the steps followed in the analyses, if possible, would contribute to the understanding and visualization of the analytical processes employed.

It is essential to clarify the use of variables in the study. This includes clarifying the mention of the "anemia level," which does not appear to have been used, and explaining whether there was a variable that represented the relationship between women or if this was developed through tests.

The authors mention that they used data from a population-based survey, but in the results, they mention the sample size. This can be confusing and should be aligned. If the research used data from the entire population, the term "sample size" is inappropriate. If a sample was used, the sample size calculation should be presented to assess the study's representativeness in the methods.

It is important to provide information about the specific tools or software used to conduct the analyses. This will help readers interested in replicating the study.

Implementing these recommendations will significantly enhance the clarity, quality, and understanding of the article, making it more accessible to a variety of audiences, including healthcare professionals and academics from various fields.

RESULTS

The authors should focus on presenting the findings and detailed descriptions of the results in the results section. Detailed methods and steps should be included in the previous section (methods). In this section (results), the main results of the analyses should be highlighted, making it easier for readers to understand what was found in each of the analyses.

It is essential for the authors to provide detailed information about the sample or population studied. This can include characteristics such as average age, gestational characteristics, residence in endemic areas, comorbidities, socioeconomic and demographic factors. These details contextualize the results and help readers understand the studied sample.

According to the journal's recommendations, table titles should be placed above the tables, not below as shown in the document. Therefore, the rules of Plos One should be reviewed, and appropriate corrections made.

If there were missing data in Table 2, these omissions should be mentioned in the article (suggested as notes in the table). Readers should be aware of any gaps or missing data.

It is important to ensure that verb tenses are correctly aligned in the text. Proper use of verb tenses helps avoid confusion and makes the text clearer. The authors should review the use of verb tenses, especially to ensure that the work is presented in the past tense, considering that the research has already been conducted.

The authors should include interpretations of the most relevant findings in the results section. This will help readers understand the significance and implications of the results. The writing focuses too much on method details, especially in the first paragraphs of the results.

In some parts of the results, the authors draw conclusions of causality. Despite the results being favorable and related to the outcomes, the apparently cross-sectional design of the study does not support causality, but rather a relationship that may be influenced by other unmeasured factors in the work. Therefore, it is suggested to review this and have the conclusions take these points into consideration, where anemic pregnant women have a higher prevalence of choosing to use bed nets.

DISCUSSION

It is essential for the authors to include a discussion section in the article. This section is crucial for comparing the findings of this work with other available evidence in the literature, highlighting where this study agrees or disagrees with works in similar or different contexts.

In the discussion section, the authors should address the study's limitations. This can include the non-consideration of relevant factors, such as incentives, professional follow-up, malaria test results, among others. Recognizing limitations helps contextualize the results and demonstrates a critical understanding of the research.

The discussion section should also highlight the practical implications of the findings. This means considering how the results can be applied in practice, in terms of public health policies, interventions, or clinical practices.

The authors can provide suggestions for future research based on the findings of this study. This helps guide future research and fill knowledge gaps.

Including a discussion section will enrich the article, making it more informative and useful for readers while demonstrating a critical analysis of the results and an acknowledgment of the study's limitations.

CONCLUSION

The conclusion should be revised to ensure that the language is accurate and concise. Ensure that all statements made in the conclusion are consistent with the study's findings and the content of the article as a whole. In the conclusion, it is mentioned that the study assessed the level of anemia, but this is not detailed, with the results only indicating whether women were anemic or not. The conclusion brings extremely relevant and interesting points to the theme; however, the work, especially the method, needs adjustments and revision to provide support for the conclusions presented. By reviewing and adjusting the conclusion of the article in accordance with these recommendations, the authors can improve the quality and clarity of the final discussion, ensuring that it aligns with the study's findings and the methodology used.

6. PLOS authors have the option to publish the peer review history of their article (what does this mean?). If published, this will include your full peer review and any attached files.

Reviewer #1: **Yes: **Ramesh Kumar

Reviewer #2: **Yes: **Elivan Silva Souza

---

## [Author Response · Author response to Decision Letter 0]

26 Dec 2023

Journal Requirements comments and Responses:

Response: Thank you for your comment. Changes have been effected in the revised manuscript.

Response: Thank you for your comment. Changes have been effected in the revised manuscript. See lines 435-422 of the revised manuscript for more information about details regarding participant consent.

Response: Thank you for your comment. Changes have been effected in the revised manuscript. See lines 208-219 of the revised manuscript for more information about patients information about written consent to have data from their medical records used in research.

Response: Thank you for your comment. Changes have been effected in the revised manuscript. See line 434 of the revised manuscript for more information.

5. Please update your submission to use the PLOS LaTeX template. 

Response:Thank you for your comment. Changes have been effected in the revised manuscript.

Reviewer #1 comments and Responses

1. Revise your title of study, this should not be a question. 

Response:Thank you for your comment. The necessary changes have been implemented in the revised manuscript. 

2. Please add statistics significance in result part of abstract 

Response: Thank you for your comment. Changes have been made to the revised manuscript. See the abstract of the revised manuscript for more information.

3. Key words should be more specific 

Response:Thank you for your comment. Changes have been made to the revised manuscript. Please see line 1 of the revised manuscript for more information.

4. Public health significance may be highlighted in the introduction part 

Response:Thank you for your comment. We have made the necessary changes in the revised manuscript. For more information, please refer to lines 3-80.

5. Conclusion needs to be revised. 

Response:Thank you for your comment. We have made the necessary changes in the revised manuscript. For more information, please refer to lines 394-424.

6. Please separate discussion part from result. 

Response:Thank you for your comment. The necessary changes have been implemented in the revised manuscript.

7. Revise references Journal title as per NLM catalogue. 

Response:Thank you for your comment. The necessary changes have been implemented in the revised manuscript.

8. Correct English grammatical error. 

Response:The Language Editing Team at the University of Energy and Natural Resources meticulously reviewed the manuscript for grammatical and spelling errors. Subsequently, we employed the Writefull Revise (https://revise.writefull.ai/) language editing software to further ensure accuracy.

Reviewer #2 comments and Responses

1. The study addresses a topic of great interest and relevance, especially in countries where malaria is endemic, such as in Africa. However, there is a clear need for substantial revisions to improve the structure of the paragraphs in order to provide greater clarity and fluency to the topic. This can be achieved by reducing repetitions of unaddressed objects and emphasizing the detailing of significant points. 

Response:Thank you for your comment. The necessary changes have been implemented in the revised manuscript.

2. Furthermore, it is important to note that the methodology of the work, while providing a detailed analysis of the statistical model formulas used, lacks crucial information. Such information includes the study design, the contexts in which the research was conducted, eligibility criteria, a description of data sources, the variables analyzed, and the treatments applied. These aspects are fundamental in epidemiological articles. 

Response:Thank you for your comment. We have made the necessary changes in the revised manuscript. For more information, please refer to lines 174-218.

3. Finally, the discussion section, which is not present in the article, needs to be incorporated. The absence of this essential element for the understanding and interpretation of the results represents a gap in the study. 

Response:Thank you for your comment. We have made the necessary changes in the revised manuscript. For more information, please refer to lines 319-371.

4. To enhance the quality of the article and meet epidemiological research standards, it is recommended to use the tools recommended by the journal Plos One, such as STROBE. 

Response:Thank you for your comment. The necessary changes have been implemented in the revised manuscript.

5. Keywords: partial least squares; Anaemic level; Social interaction; Treated bed net

Due to the importance of keywords for article discovery, it is suggested to review the proposed terms, as only "Social Interaction" is present in the DECS and MESH indexes. 

Response:Thank you for your comment. Changes have been made to the revised manuscript. Please see line 1 of the revised manuscript for more information.

6. Regarding the description of where the data can be obtained in complete sentences, a link is provided, but the identifiers (IDs) or links to the corresponding data set are not mentioned. Including this information is recommended to enable location and access. 

Response: Thank you for your comment. Changes have been effected in the revised manuscript. For more information, see line 434 in the 'Availability of Data and Materials' section of the revised manuscript.

7. Introduction: Definitions: Initially, it is essential to clearly present and define all objects of study. This includes treated bed nets, malaria, and anemia. Establishing these definitions at the beginning of the work will provide a solid foundation for the reader's understanding. 

Response:Thank you for your comment. We have made the necessary changes in the revised manuscript. For more information, please refer to lines 3-5, 20-22, and 35-38.

8. Risk Factors and Consequences: Next, I recommend presenting the consequences and associated factors related to malaria, anemia, and the non-use of bed nets in a structured manner. Although these elements have already been mentioned in various paragraphs of the introduction, structuring this information will avoid repetitions and enhance the clarity and cohesion of the text. 

Response:Thank you for your comment. We have made the necessary changes in the revised manuscript. For more information, please refer to lines 10-11, 26-34, and 38-58.

9. Outcomes and Impact: It is relevant to present, in a fluid and structured manner, the statistics related to the outcomes and consequences addressed throughout the work. This section should connect the outcomes presented in the article, providing an overview. 

Response:Thank you for your comment. We have made the necessary changes in the revised manuscript. For more information, please refer to lines 53-120.

10. Justification: It is advisable to include a justification section that demonstrates the importance of the research conducted. This involves presenting pre-existing hypotheses and explaining why the research is relevant and necessary. 

Response: Thank you for your comment. Changes have been effected in the revised manuscript. See lines 101-121 of the introduction section of the revised manuscript.

11. Regarding anemia, it is important for the introduction to provide a clearer explanation of how anemia relates to the article's theme and why it is relevant to study it. This will ensure that readers understand the importance of this variable in the research context. 

Response:Thank you for your comment. Changes have been effected in the revised manuscript. See lines 107-121 of the introduction section of the revised manuscript.

12. Finally, it is advisable to remove the paragraph that presents the structure of the article, as this is already widely known and should follow the journal's guidelines. The introduction should focus on establishing the context and relevance of the study. 

Response:Thank you for your comment. Changes have been effected in the revised manuscript.

13. It is highly recommended that the authors follow the STROBE guidelines, as suggested by Plos One. These guidelines are especially relevant for epidemiological articles, as they provide a framework to ensure the understanding and reproducibility of research in different contexts. Certainly, this will contribute to the quality and clarity of the article. 

Response:Thank you for your comment. Changes have been effected in the revised manuscript.

14. A clear description of the research design adopted in the study should be included. It appears to be an analytical cross-sectional study, but this information should be explicitly provided. Additionally, it is essential to provide the study's context, including the population size, characteristics of the country where the research was conducted, and any other details that help contextualize the work. 

Response:Thank you for your comment. We have made the necessary changes in the revised manuscript. For more information, please refer to lines 174-199.

15. Detailed information about participant eligibility criteria, exposures, outcomes, and any other relevant variables must be provided. These descriptions should be clear and included in the methods section so that readers can fully understand the scope of the study. 

Response:Thank you for your comment. We have made the necessary changes in the revised manuscript. For more information, please refer to lines 174-205.

16. The description of whether the data was anonymized or not should be in the method section, not in the results. This will contribute to a clearer organization of the article. 

Response:Thank you for your comment. Changes have been effected in the revised manuscript.

17. While it is evident that the authors have experience in statistical analysis, it is important to consider that readers may include healthcare professionals, managers, and academics who are not experts in the field. Therefore, it is recommended that mathematical formulas be provided as supplementary material, and that the authors describe the motivations behind their methodological choices, explaining how the results were processed. Additionally, creating a flowchart to present the steps followed in the analyses, if possible, would contribute to the understanding and visualization of the analytical processes employed. 

Response:Thank you for your comment. Changes have been effected in the revised manuscript.

18. It is essential to clarify the use of variables in the study. This includes clarifying the mention of the "anemia level," which does not appear to have been used, and explaining whether there was a variable that represented the relationship between women or if this was developed through tests. 

Response:Thank you for your comment. Changes have been effected in the revised manuscript.

19. The authors mention that they used data from a population-based survey, but in the results, they mention the sample size. This can be confusing and should be aligned. If the research used data from the entire population, the term "sample size" is inappropriate. If a sample was used, the sample size calculation should be presented to assess the study's representativeness in the methods. 

Response:Thank you for your comment. Changes have been made to the revised manuscript. Please see supplementary material of the revised manuscript for more information.

20. It is important to provide information about the specific tools or software used to conduct the analyses. This will help readers interested in replicating the study. 

Response:Thank you for your comment. Changes have been effected in the revised manuscript. See line 252-253 of the method section of the revised manuscript.

21. The authors should focus on presenting the findings and detailed descriptions of the results in the results section. Detailed methods and steps should be included in the previous section (methods). In this section (results), the main results of the analyses should be highlighted, making it easier for readers to understand what was found in each of the analyses. 

Response:Thank you for your comment. Changes have been effected in the revised manuscript.

22. It is essential for the authors to provide detailed information about the sample or population studied. This can include characteristics such as average age, gestational characteristics, residence in endemic areas, comorbidities, socioeconomic and demographic factors. These details contextualize the results and help readers understand the studied sample. 

Response:Thank you for your comment. Changes have been effected in the revised manuscript. See lines 172-200 of the methods section of the revised manuscript.

23. According to the journal's recommendations, table titles should be placed above the tables, not below as shown in the document. Therefore, the rules of Plos One should be reviewed, and appropriate corrections made. 

Response:Thank you for your comment. Changes have been effected in the revised manuscript.

24. If there were missing data in Table 2, these omissions should be mentioned in the article (suggested as notes in the table). Readers should be aware of any gaps or missing data. 

Response:Thank you for your comment. Changes have been effected in the revised manuscript.

25. It is important to ensure that verb tenses are correctly aligned in the text. Proper use of verb tenses helps avoid confusion and makes the text clearer. The authors should review the use of verb tenses, especially to ensure that the work is presented in the past tense, considering that the research has already been conducted. 

Response:Thank you for your comment. Changes have been effected in the revised manuscript.

26. The authors should include interpretations of the most relevant findings in the results section. This will help readers understand the significance and implications of the results. The writing focuses too much on method details, especially in the first paragraphs of the results. 

Response: Thank you for your comment. Changes have been effected in the revised manuscript. 

27. In some parts of the results, the authors draw conclusions of causality. Despite the results being favorable and related to the outcomes, the apparently cross-sectional design of the study does not support causality, but rather a relationship that may be influenced by other unmeasured factors in the work. Therefore, it is suggested to review this and have the conclusions take these points into consideration, where anemic pregnant women have a higher prevalence of choosing to use bed nets. 

Response: Thank you for your comment. Changes have been effected in the revised manuscript. See line 295 of the methods section of the revised manuscr

---

## [Editor Report · Decision Letter 1]

30 Jan 2024

PONE-D-23-14924R1The Effect of Anaemia Level on Pregnant and Non-Pregnant Women's Decision to Sleep in a Treated Bed NetPLOS ONE

Dear Dr. Kwame Ansah,

Thank you for submitting your manuscript to PLOS ONE. After careful consideration, we feel that it has merit but does not fully meet PLOS ONE’s publication criteria as it currently stands. Therefore, we invite you to submit a revised version of the manuscript that addresses the points raised during the review process.

We look forward to receiving your revised manuscript.

Kind regards,

Ibrahim Sebutu Bello, MBBS, MPH, MD, FMCGP

Academic Editor

PLOS ONE

Journal Requirements:

Additional Editor Comments:

Review Report

Observations

The authors have tried to implement the corrections suggested by the reviewers. This is commendable.

However, a few outstanding issues need to be addressed more adequately, as outlined below.

Title: The study did not assess nor include any variable called ‘anaemia level’. What was included was the presence or absence of anaemia, i.e., anaemia or not. This is not ‘anaemia level’.

Hence, I suggest the title be further modified as ‘The relationship between anaemia and the use of treated bed nets among pregnant and non-pregnant women in…. region of Ghana.’

Line 218: Authors should insert a sub-heading titled ‘Statistical Analysis.”

Line 370: The “Model Diagnostics and Validation” section, placed after the discussion, should be moved to the Methods section under the “Statistical Analysis" sub-section.

Inferences on causality.

Lines 229 – 231: This suggests that the variables ‘pregnancy status’ and ‘anaemia’ determined the decision of respondents to use treated nets. This implies a causal relationship which, in terms of health epidemiology, cannot be determined by the cross-sectional data used in the study. It should be stated simply as a relationship between the variables. The same observation applies to statements made in the discussion (lines 317-319 and 326) and conclusion (lines 392-393)

Lines 320-323: As the authors rightly observed here, the study's findings should be stated as a statistical relationship between the dependent variable (use of nets) and the covariates.

Tenses of verbs

Some tenses still need to be changed to past tests since the study has already been conducted. These include (but are not limited to) the following:

Lines 266: “We will get…”

Lines 273-274: “We will now compute…” should changed to “we then computed.”

Lines 228 – 229: “…will be used”

Conclusion

Lines 397-398: The statement here that pregnant and anaemic women had higher usage of bed nets contradicts the statement in lines 405-406, stating that bed net use among the same pregnant and anaemic women is low.

---

## [Author Response · Author response to Decision Letter 1]

20 Feb 2024

Journal Requirements: Please review your reference list to ensure that it is complete and correct. If you have cited papers that have been retracted, please include the rationale for doing so in the manuscript text, or remove these references and replace them with relevant current references. Any changes to the reference list should be mentioned in the rebuttal letter that accompanies your revised manuscript. If you need to cite a retracted article, indicate the article’s retracted status in the References list and also include a citation and full reference for the retraction notice.

Response: The reference list has been updated: Reference 47, appearing on lines 582-584, has been deleted because it was not used in the paper. This was an oversight.

Editor Comments 

1. Title: The study did not assess nor include any variable called ‘anaemia level’. What was included was the presence or absence of anaemia, i.e., anaemia or not. This is not ‘anaemia level’.

Hence, I suggest the title be further modified as ‘The relationship between anaemia and the use of treated bed nets among pregnant and non-pregnant women in…. region of Ghana.’ 

Response: Thank you for your comment. Please see the title of the revised manuscript for the updates.

2. Line 218: Authors should insert a sub-heading titled ‘Statistical Analysis.” 

Response: Thank you for your comment. Please see line 218 of the revised manuscript for the updates.

3. Line 370: The “Model Diagnostics and Validation” section, placed after the discussion, should be moved to the Methods section under the “Statistical Analysis" sub-section. 

Response: Thank you for your comment. Please see lines 314-335 of the revised manuscript for the updates.

4. Lines 229 – 231: This suggests that the variables ‘pregnancy status’ and ‘anaemia’ determined the decision of respondents to use treated nets. This implies a causal relationship which, in terms of health epidemiology, cannot be determined by the cross-sectional data used in the study. It should be stated simply as a relationship between the variables. The same observation applies to statements made in the discussion (lines 317-319 and 326) and conclusion (lines 392-393). 

Response: Thank you for your comment. Please see abstract, lines 162-165, lines 337-338, lines 393-395, and 368-369 of the revised manuscript for the updates.

5. Lines 320-323: As the authors rightly observed here, the study's findings should be stated as a statistical relationship between the dependent variable (use of nets) and the covariates. 

Response: Thank you for your comment.

6. Tenses of verbs

Some tenses still need to be changed to past tests since the study has already been conducted. These include (but are not limited to) the following:

Lines 266: “We will get…”

Lines 273-274: “We will now compute…” should changed to “we then computed.”

Lines 228 – 229: “…will be used”.

Response: Thank you for your comment. Please see lines 174-188, lines 231-238, lines 251-262, line 265, and lines 271-276 of the revised manuscript for the updates. 

7. Conclusion

Lines 397-398: The statement here that pregnant and anaemic women had higher usage of bed nets contradicts the statement in lines 405-406, stating that bed net use among the same pregnant and anaemic women is low. 

Response: Thank you for your comment. Please see line 404 of the revised manuscript for the updates.

---

## [Editor Report · Decision Letter 2]

27 Feb 2024

The relationship between anaemia and the use of treated bed nets among pregnant and non-pregnant women in Ghana

PONE-D-23-14924R2

Dear Dr. Kwame Ansah,

We’re pleased to inform you that your manuscript has been judged scientifically suitable for publication and will be formally accepted for publication once it meets all outstanding technical requirements.

Kind regards,

Ibrahim Sebutu Bello, MBBS, MPH, MD, FMCGP

Academic Editor

PLOS ONE

Additional Editor Comments (optional):

All issues raised in the review report have been satisfactorily attended to by the author(s).